# Associations between Inflammation, Hemoglobin Levels, and Coronary Artery Disease in Non-Albuminuric Subjects with and without Type 2 Diabetes Mellitus

**DOI:** 10.3390/ijms241814131

**Published:** 2023-09-15

**Authors:** Javier Donate-Correa, Ernesto Martín-Núñez, Carmen Mora-Fernández, Ainhoa González-Luis, Alberto Martín-Olivera, Juan F. Navarro-González

**Affiliations:** 1Unidad de Investigación, Hospital Universitario Nuestra Señora de Candelaria (HUNSC), 38000 Santa Cruz de Tenerife, Spain; emarnu87@gmail.com (E.M.-N.); carmenmora.fdez@gmail.com (C.M.-F.); ainhoa.gonaluz@gmail.com (A.G.-L.); olimarabe@gmail.com (A.M.-O.); 2GEENDIAB (Grupo Español para el Estudio de la Nefropatía Diabética), Sociedad Española de Nefrología, 39000 Santander, Spain; 3Instituto de Tecnologías Biomédicas, Universidad de La Laguna, 38000 Santa Cruz de Tenerife, Spain; 4RICORS2040 (RD21/0005/0013), Instituto de Salud Carlos III, 28000 Madrid, Spain; 5Servicio de Nefrología, HUNSC, 38000 Santa Cruz de Tenerife, Spain

**Keywords:** diabetes mellitus, cardiovascular risk, coronary artery disease, hemoglobin, inflammation, albuminuria

## Abstract

In this cross-sectional study, we evaluated the associations of inflammation and hemoglobin with coronary artery disease (CAD) in subjects with type 2 diabetes mellitus (T2DM) and preserved kidney function. We recruited 638 participants—254 with T2DM—subjected to coronary angiography with no known cardiovascular disease, normal glomerular filtration rates, and without albuminuria. The hemoglobin and serum levels of inflammatory markers, including high-sensitivity C-reactive protein (hs-CRP), were measured. Multivariable analyses showed that inflammatory markers were not related to the severity of the stenosis in the group of subjects with diabetes. Conversely, inflammatory cytokines and albuminuria were directly related to the percentage of stenosis in subjects without T2DM (R^2^ = 0.038, *p* < 0.001). Patients with diabetes presented lower hemoglobin levels, particularly in those who also had significant CAD (14.4 [13.6–15.1] vs. 13.6 [12.2–14.8] g/dL, *p* = 0.03). Similarly, hemoglobin levels and albuminuria were inversely related to the severity of stenosis exclusively in subjects with diabetes, even after adjusting for multiple confounding factors (R^2^ = 0.081, *p* < 0.001). We conclude that reductions in hemoglobin levels in subjects with T2DM and normoalbuminuria may constitute a more relevant risk factor for CAD than inflammation.

## 1. Introduction

Diabetes mellitus has emerged as a ruling global health challenge of the 21st century, with its prevalence skyrocketing over recent decades. The International Diabetes Federation estimates that around 537 million adults have diabetes mellitus, of which type 2 (T2DM) accounts for more than 90 percent of cases [1]. This complex metabolic disorder not only affects glucose regulation but also exerts profound and far-reaching impacts on various organ systems. Among the most alarming consequences of T2DM is its intimate association with cardiovascular disease (CVD), which is the leading cause of death in these subjects, and coronary artery disease (CAD), one of the most common macrovascular disorders [2].

The intricate relationship between T2DM and CAD represents a central focus in contemporary cardiovascular research. The evidence linking T2DM to an elevated risk of CAD is compelling, as individuals with T2DM face a two- to four-fold higher likelihood of developing CAD compared to their non-diabetic counterparts [3]. It has been described that patients with T2DM and no previous history of CAD have a similar risk for cardiac events as nondiabetic subjects with a prior myocardial infarction [4]. This increased susceptibility is rooted in the multifaceted interplay of metabolic aberrations that define T2DM, including insulin resistance, hyperglycemia, and dyslipidemia [4]. These metabolic perturbations foster the initiation and progression of atherosclerosis, the underlying pathophysiological process driving CAD development.

Atherosclerosis is characterized by the buildup of arterial plaques comprising lipids, inflammatory cells, and fibrous tissue. In individuals with T2DM, the atherosclerotic process tends to be accelerated and diffuse, affecting multiple coronary arteries [5]. This results in an increased burden of coronary artery stenosis and a heightened propensity for acute coronary events, such as myocardial infarction. Furthermore, T2DM exacerbates the adverse outcomes of CAD by impairing vascular endothelial function, promoting thrombogenesis, and amplifying the inflammatory milieu within atherosclerotic plaques [6].

Consequently, T2DM represents a significant modifier of CAD severity and prognosis. However, subsequent studies have revealed that diabetic status may not be a CVD risk factor equivalent in all conditions, highlighting the need for a multivariate approach as an adequate basis for CVD risk stratification [7,8,9]. Moreover, CAD often courses asymptomatic until severe disease develops, with the presentation of myocardial infarction or sudden cardiac death [10]. All this highlights the need for proper risk stratification in the context of CAD and T2DM, a multifaceted challenge, given the considerable heterogeneity within these patient populations. Traditional risk factors, including age, sex, smoking, and hypertension, have undoubtedly played a pivotal role in predicting CAD outcomes. However, these factors alone fail to fully elucidate the inherent variability in disease progression and severity. In recent years, there has been a growing interest in identifying novel biomarkers that can enhance the precision of risk assessment and potentially guide tailored therapeutic interventions.

One such emerging biomarker of particular interest is inflammation, a central key driver of atherosclerosis [5]. The inflammatory response is not only localized to atherosclerotic plaques but also extends systemically, resulting in elevated levels of circulating inflammatory markers, such as C-reactive protein (CRP) and interleukin-6 (IL-6) [11,12]. These markers have been implicated in the initiation and progression of atherosclerosis, playing a central role in endothelial dysfunction, plaque formation, and thrombogenesis. Higher serum levels of proinflammatory markers are present in individuals at risk for CAD [13,14,15]. The determination of these parameters as risk factors for CVD reported clear positive associations with CAD in individuals with T2DM and kidney impairment [16] but also inconsistent results in other studies that included subjects with diabetes and preserved kidney function [17].

Anemia, characterized by a deficiency in hemoglobin levels, represents yet another factor that may influence cardiovascular risk among individuals with T2DM. Hemoglobin reduction itself can exert detrimental effects on cardiovascular health by reducing oxygen-carrying capacity and increasing cardiac workload. The prevalence of anemia is notably higher in the T2DM population, primarily attributed to factors such as chronic kidney disease, iron deficiency, and inflammation [18]. However, there is scarce information about the potential implication of reduced hemoglobin levels in the development of CAD in T2DM subjects without chronic kidney disease (CKD) [19,20,21]. Although with a relatively low prevalence, anemia is also present in T2DM individuals without clinical nephropathy [18,22,23]. Moreover, kidney disease patients with diabetes tend to develop anemia earlier and to a greater degree than patients without diabetes [24]. Clinically undiagnosed early kidney disease triggering a reduction in erythropoietin production could be one of the underlying causes, particularly due to increased urinary albumin excretion. Although the association between albuminuria and hemoglobin levels in T2DM patients with preserved kidney function has been barely studied, a few existing studies show increased albuminuria in patients with anemia [18,25]. Albuminuria is a non-traditional risk factor linked to the development of CVD in subjects with T2DM [26] that doubles the risk of cardiovascular death in these patients [27]. Thus, it is essential to consider the interplay between anemia, inflammation, renal function, and CAD, as these factors may be intertwined in a complex web of pathophysiological mechanisms.

On the other hand, many different substrates have been proposed as mediators of the intricate relationship between diabetes, inflammation, and cardiovascular risk. In recent years, the role of neural substrates has been increasingly recognized, and different studies evidence the pivotal roles of neural pathways, such as the autonomic nervous system and the hypothalamic–pituitary–adrenal axis, in regulating systemic inflammation, glucose metabolism, and vascular function [28]. This is the case for metabolic pathways such as the tryptophan–kynurenine system, which is altered in certain neurological disorders and may influence the inflammatory response in patients with T2DM and thus indirectly modulate the risk of CAD [29]. Furthermore, a clear link has been demonstrated between impaired neural responses and the regulation of cardiac function, which may have implications between the presence of mental health problems, increasingly recognized in the modern world as a health risk factor, and the modulation of physiological processes important in cardiovascular risk [30,31,32]. Thus, the dysregulation of these pathways in diabetes could contribute to chronic inflammation and endothelial dysfunction, key drivers of cardiovascular complications.

In this paper, we aim to explore the intricate associations between inflammation, hemoglobin levels, and CAD in two large populations of non-albuminuric subjects with and without T2DM and no previous cardiovascular events. We aspire to provide valuable insights that may promote the development of more precise diagnostic tools and therapeutic strategies for this high-risk patient population.

## 2. Results

The demographic, clinical, and biochemical data of subjects with and without T2DM are shown in Table 1. Eight hundred and fifty-one patients were considered for enrolment in the study. A total of 213 were excluded due to the exclusion criteria. Therefore, 638 patients (446 male) were finally included (Table 1). Two hundred and fifty-four subjects had T2DM. The mean age was 65.2 ± 11.3 years, and the median value of body mass index (BMI) was 26.5 ± 5.1 kg/m^2^. There were no differences in demographics and comorbidities between the two groups. Significant CAD was present in 305 (47.8%) patients, with a similar prevalence in diabetic and non-diabetic subjects (45.9% vs. 46.6%, *p* = 0.11, respectively). The SSI also presented similar median values in both groups (32% [16.3–51.4] vs. 34.4% [16.1–55], *p* = 0.64). Anemia was present in 123 patients (19.3%), with a trend of being more prevalent in the group of subjects with T2DM (22.7% vs. 16.9%, *p* = 0.08). However, hemoglobin levels were slightly but significantly reduced in the group of patients with diabetes (14 [13.1–15] vs. 13.8 [12.4–14.9] g/dL, *p* = 0.039). Serum total cholesterol, HDL- and LDL-cholesterol levels were lower in the subjects with T2DM, probably due to the higher percentage of patients treated with lipid-lowering medications in this group (23.7% vs. 45.7%, *p* = 0.03).

Fasting glucose and Hb1Ac were higher in the group with T2DM (98 [89–112.8] vs. 148 [133–183.3] mg/dL, *p* < 0.001; 5.6 ± 0.97 vs. 7.8 ± 0.89%, *p* < 0.001, respectively). Similarly, the TyG index also presented significantly higher values in this group of patients (4.77 [4.63–4.93] vs. 4.93 [4.76–5.17], *p* < 0.001). No differences between groups were observed in the mean eGFR (99.1 ± 7.1 vs. 96.3 ± 9.2 mL/min/1.73 m2, *p* = 0.11). Median ACR values were higher in the group of subjects with T2DM, a difference that almost reached statistical significance (6.4 [3.6–14.6] vs. 8.37 [4.1–16] mg/g, *p* = 0.08). Of the different inflammatory parameters analyzed, including inflammatory cytokines, hs-PCR, and NLR, only TNFα differed between groups, presenting higher values in the group of patients with diabetes (1.99 [1.38–2.67] vs. 2.2 [1.59–2.99] pg/mL, *p* < 0.01).

We performed bivariate correlation analyses in each group to determine the associations of hemoglobin levels with different parameters (Table 2). Hemoglobin concentration was inversely related with ACR both in non-T2DM (r = −0.103, *p* = 0.043) and T2DM patients (r = −0.224, *p* = 0.005). Hemoglobin was also significantly and inversely associated with fasting glucose (r = −0.199, *p* = 0.013) and hs-CRP values (r = −0.167, *p* = 0.038) only in the group of subjects with T2DM. Interestingly, and again exclusively in T2DM patients, hemoglobin was inversely related to the values of the SSI (r = −0.19, *p* = 0.018), particularly with the percentage of stenosis observed in two major epicardial arteries: RCA (r = −0.167, *p* = 0.039) and LAD (r = −0.182, *p* = 0.036) (Table 2).

Among the group of subjects with T2DM, those with anemia presented higher values of SSI (52.8 [31.5–56.8] vs. 29 [14.5–52.8], *p* = 0.0035) (Figure 1). This difference, according to the anemia condition, was absent both in the whole population (33.9 [17.1–53.9] vs. 32.3 [16.3–53.8], *p* = 0.464) and in the group without diabetes (27.5 [15.5–42.8] vs. 33 [16.7–54], *p* = 0.192).

The sub-analysis of the group of subjects with T2DM attending to the occurrence of significant CAD revealed a higher incidence of anemia in patients with significant CAD (13.1% vs. 34.2%, *p* = 0.03) (Table 3). Congruently, the subjects with CAD also presented reduced levels of hemoglobin (14.4 [13.6–15.1] vs. 13.6 [12.2–14.8] g/dL, *p* = 0.03), together with increased levels of ACR (4.73 [3.25–12.6] vs. 9 [4.4–18.8] mg/g, *p* = 0.04) and hs-CRP (3.14 [1.3–6.1] vs. 4.6 [2.1–6.8] mg/L, *p* = 0.02), and higher SBP (*p* = 0.04). No differences were observed in TyG index values (5 [4.74–5.34] vs. 4.93 [4.77–5.14], *p* = 0.36), nor in inflammatory cytokines TNFα and IL6 (2 [1.59–2.78] vs. 2.22 [1.59–3.1] pg/mL, *p* = 0.19; 5.43 [3–8.89] vs. 7.47 (3.51–14) pg/mL, *p* = 0.21, respectively), nor in NLR values (2.51 [1.75–3.68] vs. 2.19 [1.74–3.38], *p* = 0.46).

The association between hemoglobin levels and SSI in T2DM was also observed after additional adjustment for potential confounders: age, sex, HT, current smokers, dyslipidemia, ACR, BMI, phosphorus, TNFα, IL6, and hs-CRP. Thus, the results of a forward stepwise multiple regression analysis performed with the SSI as the dependent variable showed that reduced hemoglobin levels and ACR were independently related to the degree of stenosis severity (adjusted R^2^ = 0.081, *p* < 0.001) (Table 4). Again, this association was not present in the group of non-T2DM subjects, where TNFα and IL6, together with ACR, were the variables significantly and independently associated with the SSI (adjusted R^2^ = 0.045, *p* < 0.05).

## 3. Discussion

The results of this study indicate that inflammation and reduced hemoglobin levels are factors differentially associated with CAD in subjects with and without T2DM and normal kidney function (Figure 2). Thus, patients with diabetes and anemia presented a higher incidence of significant CAD than those with normal hemoglobin levels. Moreover, hemoglobin levels in T2DM subjects were inversely and independently related to the degree of coronary stenosis. This association between hemoglobin and CAD was absent in a comparative group of subjects without diabetes with similar age, sex distribution, BMI, and comorbidities, including significant CAD and kidney function. In this group, inflammatory cytokines TNFα and IL6 were the main factors directly related to the severity of coronary stenosis. In both groups of patients, the ACR was a factor positively associated with the degree of stenosis. Given the prevalence of coronary heart disease in subjects with T2DM, the early detection and management of reduced hemoglobin levels could constitute a relevant objective to potentially prevent or reverse these complications, whereas, in non-diabetic subjects, the modulation of inflammation may be an earlier therapeutic target.

The excess of CVD in T2DM patients, when compared to non-diabetic subjects, is attributed to a higher prevalence of traditional risk factors, including dyslipidemia, hypertension, and obesity. Although an adequate control of these well-known risk factors has been shown to be of great importance in reducing the risk of CVD in chronic patients [33,34], other non-traditional risk factors may also be relevant [35]. Very few studies have prospectively demonstrated the independent role of these non-traditional risk factors, including inflammation, increased albuminuria, and anemia in subjects with T2DM [36,37]. In addition, it has also been proposed that some of these variables can be specific risk factors for CVD in diabetes, which could help to explain the increased morbidity observed in T2DM [37].

Anemia is a non-traditional risk factor for CVD that represents an emerging global health problem and requires a significant allocation of healthcare resources. The signs and symptoms of anemia vary with severity and the period over which the reduction in hemoglobin develops. Undetected and, therefore, untreated anemia leads to altered menstrual cycles, anorexia, depression, cognitive dysfunction, decreased libido, reduced exercise capacity, fatigue, and consequently impaired quality of life [38]. The association of anemia with CVD has been described in several studies that have supported the idea that this association is mainly seen in patients with CKD [39,40,41]. Thus, most of them included clinical populations with advanced kidney failure [42], end-stage renal disease [43], or diabetic nephropathy [18,25]. However, although often comorbid with chronic diseases, anemia alone has a nearly identical association with mortality as CKD [44]. Our results indicate an independent association between reduced hemoglobin concentrations and the prevalence and severity of CAD in T2DM with preserved kidney function. Therefore, this also points to the suitability of early detection of anemia in these patients in order to be treated once diagnosed since it may contribute to the pathogenesis and progression of this complication. Thus, regular screening for anemia, along with other complications associated with diabetes, could help slow the progression of cardiovascular complications in these patients [45].

The prevalence of anemia in diabetes mellitus in the absence of kidney disease has been demonstrated in previous studies [25,45]. However, very few studies explored the association between albuminuria and the prevalence of anemia in the subjects with preserved kidney function in the general population [46,47] and none, to our knowledge, in T2DM patients. In our study, the hemoglobin levels were inversely related to albuminuria but were independent of eGFR values, suggesting that erythropoietin renal deficiency begins even before there is evidence of deterioration in kidney function, i.e., albuminuria. The underlying mechanisms of this association are not clear but are probably debt to the link between albuminuria and proximal tubule injury [48]. Thus, the function of interstitial fibroblasts producing erythropoietin may be impaired in this process. In addition, albuminuria is associated with low-grade inflammation [49], a condition tightly linked to anemia [50]. Diabetes mellitus per se is a chronic inflammatory state characterized by increased levels of proinflammatory cytokines [51]. This is in agreement with the results of our study, where hemoglobin levels were inversely related to ACR values in both groups of subjects, with and without diabetes and with hs-CRP concentration only in the patients with T2DM.

In conclusion, we found a clear association between anemia and CAD in normoalbuminuric patients with T2DM. Due to its insidious onset, anemia is often asymptomatic and only picked up on routine blood analysis. Early detection and correction of anemia in diabetes is important as the evidence already shows that it occurs earlier [24] and with greater severity [52] than in the non-diabetic population in the course of kidney deterioration. While patients with T2DM and microalbuminuria are increasingly being identified and treated, screening for anemia should, perhaps, be extended to this high-risk population independently of ACR values. Early corrections of anemia in T2DM subjects with early signs of nephropathy may potentially prevent or reverse these complications. We acknowledge several limitations. This study involved a relatively small sample size, so the findings may not be generalizable to a broader community. Although we accounted for the confounding of traditional cardiovascular risk factors, a potential for uncontrolled or residual confounding that could affect the hemoglobin–coronary stenosis relationship is plausible. Finally, given the cross-sectional design of this study, we can only demonstrate associations without definitive inferences on their direction or causality. Nevertheless, this is the first study linking reductions in hemoglobin levels with the degree of stenosis severity in subjects with T2DM and normal kidney function. Further experimental and clinical studies are warranted to confirm our findings.

## 4. Materials and Methods

### 4.1. Study Design and Population

Cross-sectional study designed to evaluate the association of hemoglobin and inflammatory parameters with the occurrence and severity of CAD in subjects with and without T2DM. The population included consecutive patients undergoing nonemergency diagnostic evaluation for CAD via elective coronary angiography. The inclusion criteria were age > 18 years old, T2DM duration of at least 1 year in diabetic subjects, no evidence of previous cardiovascular disease, no CKD defined as normal kidney function (estimated glomerular filtration rate (eGFR) > 60 mL/min/1.73 m^2^) and normoalbuminuria (urine albumin-to-creatinine ratio (ACR) < 30 mg/g). Exclusion criteria included previous myocardial infarction, coronary angioplasty, intracoronary stent placement or coronary artery bypass graft surgery, stroke or transient ischemic attack, peripheral vascular disease, hemodynamic instability, cardiac arrhythmia, immunologic or inflammatory diseases (including rheumatoid arthritis, systemic lupus erythematosus, or inflammatory bowel disease). In addition, we also excluded non-diabetic subjects with fasting glucose and HbA1c values upper to 100 mg/dL and 5.7%, respectively. No patient was receiving calcium, phosphate, or vitamin D supplementation. All the protocols complied with the ethical standards of the Declaration of Helsinki and were reviewed and approved by the institutional ethics committee. Written informed consent was obtained from all participants.

### 4.2. Coronary Angiography

Coronary angiography was performed using standard techniques. The assessment of coronary stenosis was determined in four major epicardial arteries: left main coronary artery (LCA), left anterior descending artery (LAD), circumflex artery (CA), and right coronary artery (RCA). A stenosis severity index (SSI) was defined as the average of the maximum stenosis in each of those arteries. Significant CAD was defined as the presence of at least one lesion leading to ≥50% lumen diameter stenosis in any of the considered arteries.

### 4.3. Clinical and Biochemical Variables

Demographic and clinical data were obtained from each participant. General biochemical analyses were performed using standard laboratory methods. Biochemical variables were determined in fasting blood and urine samples before angiography. We evaluated systemic inflammation by determining the serum levels of the inflammatory cytokines tumor necrosis factor (TNF) α and interleukin (IL) 6, and also of high-sensitivity C-reactive protein (hs-CRP) and the neutrophil-to-lymphocyte ratio (NLR). Cytokines TNFα and IL6 were measured using high-sensitivity ELISA methods (Quantikine^®^, R&D Systems, Abingdon, UK). Minimum detectable concentrations were 0.10 pg/mL and 0.50 pg/mL, respectively. Intra- and inter-assay coefficients of variability were <10.8%. hs-CRP was measured using a high-sensitivity particle-enhanced immunoturbidimetric fully automated assay in a Cobas 6000 analyzer (Roche Diagnostics GmbH, Mannheim, Germany) with a sensitivity of 0.3 mg/L and intra- and inter-assay coefficients of variation of 1.6% and 8.4%, respectively. NLR was calculated for each patient by dividing the absolute number of neutrophils by the absolute number of lymphocytes. We also determined the triglyceride-glucose (TyG) index, calculated as TyG index = ln [Fasting triglyceride (mg/dL) × fasting glucose (mg/dL)]/2, as a potential marker of vascular dysfunction. Patients were considered to have anemia when presenting hemoglobin <13 g/dL (men) or <12 g/dL (women).

### 4.4. Statistical Analysis

Quantitative variables were presented as mean and standard deviation or median and interquartile intervals, and categorical data were expressed as frequencies and percentages. Comparisons between groups were performed using Chi-square test, Student’s t-test, or Mann–Whitney U test as appropriate. The Spearman correlation coefficient was calculated to assess the relation between soluble hemoglobin concentrations and other variables, including those obtained in coronary angiography study. Forward stepwise multiple regression analysis was performed to determine the independent association between potential predictor variables and the severity of CAD expressed as SSI. Collinearity was excluded by examining tolerance and the variance inflation factor (VIF) for each variable in the regression. All analyses were performed using SPSS software version 25 (IBM Corp. Armonk, NY, USA). A 2-tailed *p*-value less than 0.05 was considered statistically significant.

## 5. Conclusions

A clear association between anemia and CAD is present in patients with T2DM and normoalbuminuria. While microalbuminuria in patients with T2DM has increasingly been identified and treated, the screening for anemia should, perhaps, be extended to this high-risk population independently of ACR values. Our results suggest that the early detection and correction of anemia in routine blood tests in this population with early signs of nephropathy can potentially improve the progression of this complication.

## Figures and Tables

**Figure 1 ijms-24-14131-f001:**
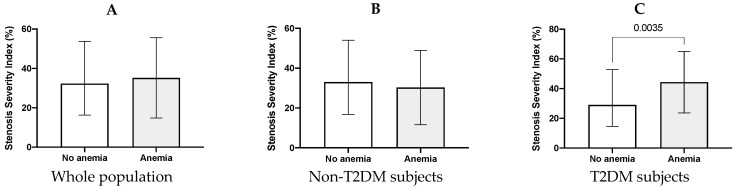
(**A**) SSI values levels in the whole population according to the presence of anemia. (**B**) and (**C**) SSI according to the presence of anemia in patients with and without T2DM, respectively. Data are presented as median and interquartile range (IQR). T2DM, type 2 diabetes mellitus; SSI, stenosis severity index.

**Figure 2 ijms-24-14131-f002:**
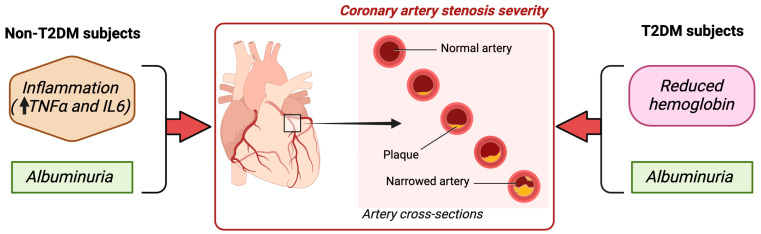
Our results indicate that in addition to albuminuria, inflammation, and reduced hemoglobin levels are risk factors differentially associated with CAD in subjects with and without T2DM and normal kidney function. ↑: upregulation.

**Table 1 ijms-24-14131-t001:** Demographic and biochemical characteristics of patients included in this study.

	Non-T2DM	T2DM	*p*	All Subjects
Characteristics				
N	384	254		638
Age (years)	66 (56–75)	65.5 (58–73)	0.99	65.2 ± 11.3
Sex (% male)	276 (71.9)	170 (66.9)	0.15	446 (69.9)
BMI (kg/m^2^)	24.8 ± 3.1	27.2 ± 5.2	0.1	26.5 ± 5.1
SBP (mm Hg)	124 ± 12.1	129 ± 10.1	0.21	126 ± 10.1
DBP (mm Hg)	74.6 ± 7.2	75.7 ± 8.6	0.17	74.9 ± 7.2
Comorbidities				
Anemia (%)	65 (16.9)	58 (22.7)	0.08	123 (19.3)
Significant CAD (%)	178 (46.6)	117 (45.9)	0.11	305 (47.8)
Obesity (%)	57 (14.8)	46 (18.1)	0.11	83 (13)
Hypertension (%)	107 (27.9)	73 (28.6)	0.48	180 (28.2)
Former smokers (%)	83 (21.6)	53 (20.1)	0.24	114 (21.3)
Current smokers (%)	118 (30.7)	59 (23.4)	0.14	177 (27.7)
Dyslipidemia (%)	178 (46.4)	107 (42.2)	0.22	285 (44.7)
Laboratory data				
SSI (%)	32 (16.3–51.4)	34.4 (16.1–55)	0.64	32.4 (16.3–53.8)
Hb (g/dL)	14 (13.1–15)	13.8 (12.4–14.9)	0.039	13.9 (12.9–15)
T-cholesterol (mg/dL)	173.3 (149–200)	168 (139.8–189)	0.04	173.3 (145–197.3)
HDL-C (mg/dL)	41 (33–46.8)	37 (30–43.3)	0.02	39 (32–46)
LDL-C (mg/dL)	100.9 (83–125)	95 (75–114.3)	<0.01	100.9 (80–122)
TG (mg/dL)	140 (102–167)	140 (111.8–198.5)	0.17	140 (105.5–173.5)
FG (mg/dL)	98 (89–112.8)	148 (133–183.3)	<0.001	119.7 ± 47
Hb1ac (%)	5.6 ± 0.97	7.8 ± 0.89	<0.001	6.37 ± 1.62
TyG index	4.77 (4.63–4.93)	4.93 (4.76–5.17)	<0.001	4.82 (4.65–4.99)
eGFR (mL/min/1.73 m^2^)	99.1 ± 7.1	96.3 ± 9.2	0.11	97.3 ± 6.2
Creatinine (mg/dL)	0.91 (0.75–1.04)	0.87 (0.71–1.03)	0.13	0.9 (0.74–1.04)
ACR (mg/g)	6.4 (3.6–14.6)	8.37 (4.1–16)	0.08	6.9 (3.6–15)
Uric acid (mg/dL)	5.8 (4.7–6.7)	5.4 (4.5–6.4)	0.23	5.7 (4.6–6.7)
Leukocytes	8 (6.6–9.4)	7.8 (6.5–9.5)	0.97	7.9 (6.5–9.4)
Monocytes	5.9 (0.7–8.8)	5.5 (0.7–8.4)	0.35	5.8 (0.7–8.8)
Lymphocytes	15.6 (2.1–26.3)	13.7 (2.1–26.8)	0.88	14.7 (2.1–26.3)
Neutrophyles	49 (5.1–63)	44.1 (4.9–64.4)	0.92	48.3 (5–63.2)
NLR (/mL)	2.46 (1.75–3.36)	2.29 (1.74–3.41)	0.85	2.42 (1.74–3.27)
hs-CRP (mg/L)	3.5 (2–6.8)	3.8 (2–6.72)	0.77	3.6 (2–6.8)
TNFα (pg/mL)	1.99 (1.38–2.67)	2.2 (1.59–2.99)	<0.01	2.05 (1.45–2.78)
IL6 (pg/mL)	6.55 (3.5–11.5)	6.5 (3.5–12.9)	0.43	6.54 (3.55–11.93)
Medication				
Statin (%)	91 (23.7)	116 (45.7)	0.03	207 (32.5)
ACEI/ARB (%)	78 (20.3)	81 (31.9)	0.12	159 (24.9)

T2DM, type 2 diabetes mellitus; BMI, body mass index; SBP, systolic blood pressure; DBP, diastolic blood pressure; CAD, coronary artery disease; HDL-C high-density lipoprotein cholesterol; LDL-C low-density lipoprotein cholesterol; TG, triglycerides; TyG, triglyceride glucose index; FG, fasting glucose; Hb1ac, glycated hemoglobin; eGFR, estimated glomerular filtrate rate; ACR, urine albumin-to-creatinine ratio; hs-CRP high sensitivity C-reactive protein, TNFα, tumor necrosis factor alpha; IL, interleukin; NLR, neutrophil/lymphocyte ratio; ACEI, angiotensin-converting enzyme inhibitor; ARB, angiotensin II receptor antagonist.

**Table 2 ijms-24-14131-t002:** Bivariate correlation of hemoglobin levels with ACR, glucose, inflammatory markers, and coronary artery stenosis determinations in subjects with and without T2DM.

	Non-T2DM (n = 384)		T2DM (n = 254)	
	r	*p*	r	*p*
ACR (mg/g)	−0.103	0.043	−0.224	0.005
FG (mg/dL)	−0.036	0.48	−0.199	0.013
hs-CRP (mg/L)	−0.004	0.943	−0.167	0.038
TNFα (pg/mL)	−0.13	0.8	−0.01	0.98
IL6 (pg/mL)	−0.078	0.13	−0.069	0.39
NLR	−0.041	0.426	−0.181	0.25
SSI	−0.03	0.948	−0.19	0.018
Obs LCA (%)	0.009	0.863	−0.055	0.5
Obs RCA (%)	−0.331	0.551	−0.167	0.039
Obs LAD (%)	0.017	0.734	−0.175	0.036
Obs CA (%)	0.06	0.913	−0.014	0.861

T2DM, type 2 diabetes mellitus; ACR, urine albumin-to-creatinine ratio; hs-CRP high sensitivity C-reactive protein, TNFα, tumor necrosis factor alpha; IL, interleukin; NLR, neutrophil/lymphocyte ratio; SSI, severity stenosis index; Obs, obstruction; LCA, left main coronary artery; RCA, right coronary artery; LAD, left anterior descending artery; CA, circumflex artery.

**Table 3 ijms-24-14131-t003:** Sub-analysis of demographic and biochemical characteristics of subjects with T2DM according to the presence of significant coronary artery disease.

	Non-CAD	CAD	*p*
Characteristics			
N	137	117	
Age (years)	61 (54–73)	66 (59–74)	0.22
Sex (% male)	90 (65.7)	80 (68.4)	0.39
BMI (kg/m^2^)	26.8 ± 2.2	27.3 ± 2.9	0.13
SBP (mm Hg)	129 ± 10.1	131 ± 11.1	0.04
DBP (mm Hg)	75.3 ± 7.7	75.9 ± 9.1	0.47
Comorbidities			
Anemia (%)	18 (13.1)	40 (34.2)	0.03
Obesity (%)	24 (17.5)	22 (18.8)	0.74
Hypertension (%)	33 (24.1)	40 (34.2)	0.51
Former smokers (%)	25 (18.3)	28 (23.9)	0.62
Current smokers (%)	32 (23.4)	27 (23.1)	0.47
Dyslipidemia (%)	51 (37.2)	56 (47.9)	0.69
Laboratory data			
Hb (g/dL)	14.4 (13.6–15.1)	13.6 (12.2–14.8)	0.03
T-cholesterol (mg/dL)	162 (136–192)	168 (140–188)	0.96
HDL-C (mg/dL)	33 (25–41.7)	37 (31–44)	0.17
LDL-C (mg/dL)	96 (82–120)	94 (74–111)	0.44
TG (mg/dL)	129 (107–198)	141 (112–200)	0.63
FG (mg/dL)	135 (127–165)	153 (134–188)	0.09
Hb1ac (%)	7.42 ± 0.2	7.58 ± 0.21	0.11
TyG index	5 (4.74–5.34)	4.93 (4.77.5.14)	0.36
eGFR (ml/min/1.73 m^2^)	99.8 ± 10.3	96.9 ± 9.3	0.16
Creatinine (mg/dL)	0.9 (0.79–1.05)	0.86 (0.71–1.02)	0.28
ACR (mg/g)	4.73 (3.25–12.6)	9 (4.4–18.8)	0.04
Uric acid (mg/dL)	5.54 (4.46–6.79)	5.42 (4.51–6.23)	0.84
Leukocytes	7.7 (6.4–9.3)	7.8 (6.5–9.6)	0.98
Monocytes	7.8 (0.8–9)	7.2 (0.6–8.4)	0.04
Lymphocytes	18.8 (2.9–27.2)	3.3 (2–25.9)	0.11
Neutrophyles	52.9 (6.8–67.1)	49.3 (4.7–62.1)	0.44
NLR	2.51 (1.75–3.68)	2.19 (1.74–3.38)	0.46
hs-CRP (mg/L)	3.14 (1.3–6.1)	4.6 (2.1–6.8)	0.02
TNFα (pg/mL)	2 (1.59–2.78)	2.22 (1.59–3.1)	0.19
IL6 (pg/mL)	5.43 (3–8.89)	7.47 (3.51–14)	0.21
Medication			
Statin (%)	61 (44.5)	55 (47.2)	0.24
ACEI/ARB (%)	37 (27)	44 (37.4)	0.31

CAD, coronary artery disease; BMI, body mass index; SBP, systolic blood pressure; DBP, diastolic blood pressure; HDL-C high-density lipoprotein cholesterol; LDL-C low-density lipoprotein cholesterol; TG, triglycerides; TyG, triglyceride glucose index; FG, fasting glucose; Hb1ac, glycated hemoglobin; eGFR, estimated glomerular filtrate rate; ACR, urine albumin-to-creatinine ratio; hs-CRP high sensitivity C-reactive protein, TNFα, tumor necrosis factor alpha; IL, interleukin; NLR, neutrophil/lymphocyte ratio; ACEI, angiotensin-converting enzyme inhibitor; ARB, angiotensin II receptor antagonist.

**Table 4 ijms-24-14131-t004:** Multiple stepwise regression analysis for stenosis severity index as the dependent variable in T2DM and non-T2DM subjects.

Stenosis Severity Index	Adjusted R^2^	ß	SE	t	*p*
Non-T2DM subjects	0.038				<0.001
ACR (mg/g)		0.447	0.138	3.234	0.001
IL6 (pg/mL)		0.094	0.052	1.815	0.03
TNFα (pg/mL)		0.349	0.205	1.702	0.04
T2DM subjects	0.081				<0.001
Hb (g/dL)		−1.705	0.939	−1.816	0.01
ACR (mg/g)		0.695	0.215	3.238	0.001

T2DM, type 2 diabetes mellitus; TNFα, tumor necrosis factor alpha; ACR, urine albumin-to-creatinine ratio.

## Data Availability

Proposals relating to data access should be directed to the corresponding authors.

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
