# Peer review of "Associations between Inflammation, Hemoglobin Levels, and Coronary Artery Disease in Non-Albuminuric Subjects with and without Type 2 Diabetes Mellitus"

_ijms, 2023, doi:10.3390/ijms241814131_

Round 1

Reviewer 1 Report

In this work entitled " Inflammation and reduced hemoglobin levels are differentially associated with coronary artery disease in non-albuminuric subjects with and without type 2 diabetes mellitus” aims to evaluate the clinical significance of inflammation and hemoglobin in relation to the risk of coronary artery disease in subjects with type 2 diabetes mellitus (T2DM) and preserved kidney function. This is cross-sectional study, with recruited 638 participants -254 with T2DM- with no known cardiovascular disease, normal glomerular filtration rates and without albuminuria that were subjected to coronary angiography. This is very interesting. The paper is well written.

-           How to compare with published data?

-           What authors propose for future trends?

Author Response

Dear Reviewer 1. 

Tanks for your revision and your kind words.

  • How to compare with published data?

The comparison with previous studies is complex. Although the prevalence of anemia in diabetes mellitus in the absence of kidney disease has been demonstrated in previous studies [Diabetes Res Clin Pract. 2009;85:179–182; Kidney Int. 2005; 67:1483–1488], to our knowledge, none have explored the association between albuminuria and the prevalence of anemia in subjects with preserved renal function in patients with DM2. In our study, we suggest that renal erythropoietin deficiency begins even before there is evidence of renal failure, ie albuminuria. There are studies that can help us explain this observation; therefore, a link has been established between albuminuria and proximal tubule damage, which can alter the function of interstitial fibroblasts that produce erythropoietin [Nephrol Dial Transplant. 2016;31:1460–1470]. Furthermore, albuminuria is associated with low-grade inflammation [Clin J Am Soc Nephrol. 2012;7:1938–1946], a condition closely related to anemia [N Engl J Med. 2005;352:1011–1023]. Diabetes mellitus per se is a chronic inflammatory state characterized by elevated levels of proinflammatory cytokines [Nephrol. dial. transplant. 17 (Suppl. 11), 39–43 (2002)]. This agrees with the results of our study, where the hemoglobin levels were inversely related to the ACR values in both groups of subjects -with and without diabetes- and with the concentration of PCRas only in patients with DM2.
All these reasoning and references are included in the discussion section.

  • What authors propose for future trends?

Probably the most applicable conclusion in clinical practice for the treatment of diabetes is the need for early detection of anemia in the course of deteriorating renal function. With this idea in mind, clinical studies would be necessary to verify whether early correction of anemia in subjects with DM2 with early signs of nephropathy can potentially prevent or attenuate the development of coronary stenosis.
As we commented in the manuscript, early detection and correction of anemia in diabetes is important since the evidence already shows that it occurs earlier and with greater severity than in the non-diabetic population in the course of renal deterioration. Although more and more patients with DM2 and microalbuminuria are being identified and treated, anemia screening should perhaps be extended to this high-risk population regardless of ACR values.

Reviewer 2 Report

Donate-Correa and colleagues in the present research article entitled ‘Inflammation and reduced hemoglobin levels are differentially associated with coronary artery disease in non-albuminuric subjects with and without type 2 diabetes mellitus’ investigate the relationships between inflammation, hemoglobin levels, and coronary artery disease (CAD) in individuals without kidney damage and both with and without type 2 diabetes mellitus (T2DM). The authors highlight the significance of CAD as a leading cause of death in people with diabetes and the need for improved risk stratification methods. They emphasize the potential of non-traditional risk factors, such as inflammation and anemia, in contributing to CAD in these populations. The study's cross-sectional design involved patients undergoing elective coronary angiography. The inclusion criteria comprised age, T2DM duration, absence of previous cardiovascular disease, normal kidney function, and normoalbuminuria. The researchers excluded those with certain medical conditions. Data were collected on demographic, clinical, and biochemical variables, including inflammatory markers and hemoglobin levels. In the results section, the authors present demographic and clinical data, comparing individuals with and without T2DM. They explore the associations between hemoglobin levels, inflammatory markers, and CAD. Notably, they find that reduced hemoglobin levels are linked with higher rates of significant CAD in patients with T2DM. Additionally, in T2DM patients, hemoglobin levels were inversely related to the severity of coronary stenosis. The discussion section delves into the implications of the findings. The authors emphasize that their results suggest that reduced hemoglobin levels and inflammation play differing roles in CAD development among individuals with and without T2DM. They propose that early detection and management of anemia could be crucial in preventing or mitigating cardiovascular complications in T2DM patients. The discussion also acknowledges the study's limitations, including its sample size and cross-sectional design, while suggesting avenues for future research. In conclusion, the paper sheds light on the complex associations between inflammation, hemoglobin levels, and CAD in individuals with and without T2DM. It highlights the potential clinical relevance of identifying and addressing reduced hemoglobin levels as well as inflammation in these populations to mitigate cardiovascular risks.

In general, I think the idea of this article is really interesting and the authors’ fascinating observations on this timely topic may be of interest to the readers of International Journal of Molecular Sciences. However, some comments, as well as some crucial evidence that should be included to support the author’s argumentation, needed to be addressed to improve the quality of the manuscript, its adequacy, and its readability prior to the publication in the present form.

Please consider the following comments:

I recommend revising the title. The title's structure is quite complex, which might make it harder for readers to quickly grasp the main message, therefore simplifying the structure could enhance readability. Also, in my opinion, in the present form it is quite long, which might discourage some readers from engaging with the article. A potential revised title could be: "Associations Between Inflammation, Hemoglobin Levels, and Coronary Artery Disease in Non-Albuminuric Individuals With and Without Type 2 Diabetes." [1-3].

A graphical abstract that will visually summarize the main findings of the manuscript is highly recommended.

Abstract: According to the Journal’s guidelines, this section should be presented as a short summary of about 200 words maximum that objectively represents the article. It should let readers get the gist or essence of the manuscript quickly, prepare the readers to follow the detailed information, analyses, and arguments in the full paper and, most of all, it should help readers remember key points from your paper. Please, consider rewrite this paragraph following these instructions [4]. 

Keywords: Please list ten keywords chosen from Medical Subject Headings (MeSH) and use as many as possible in the title and in the first two sentences of the abstract. I would suggest adding “Cardiovascular Risk” as keyword.

Introduction: The authors need to reorganize this section with several paragraphs made up of about 1000 words, introducing information on the main constructs of this study, which should be understood by a reader in any discipline, and making it persuasive enough to put forward the main purpose of the current research the author has conducted and the specific purpose the author has intended by this protocol. I would like to encourage the authors to present the introduction starting with the general background, proceeding to the specific background on the prevalence of T2DM, its association with cardiovascular disease, the complexity of risk stratification, the role of biomarkers, the connection between inflammation and atherosclerosis, the potential role of anemia in cardiovascular risk among T2DM patients. Those main structures should be organized in a logical and cohesive manner [5]. 

In this regard, I believe that the Introduction section would benefit from additional information to enhance its clarity and contextualization. To strengthen this section, I suggest incorporating a brief discussion about the potential neural substrates that connect diabetes, inflammation, and cardiovascular risk. Recent research has increasingly recognized the role of neural substrates in mediating the intricate relationship between diabetes, inflammation, and cardiovascular health. Neural pathways, such as the autonomic nervous system and the hypothalamic-pituitary-adrenal axis, play pivotal roles in regulating systemic inflammation, glucose metabolism, and vascular function. The dysregulation of these pathways in diabetes could contribute to chronic inflammation and endothelial dysfunction, key drivers of cardiovascular complications. Exploring the neural underpinnings of the observed associations between reduced hemoglobin levels, inflammatory markers, and CAD in individuals with T2DM could provide valuable insights into the complex interplay of physiological processes involved [6-7]. Addressing this aspect could potentially open avenues for more targeted interventions that focus on modulating neural substrates alongside conventional risk factors. By integrating this discussion of neural substrates, the manuscript would not only underscore the multidimensional nature of the investigated associations but also foster a deeper understanding of the mechanisms at play [8-10]. This addition would further contribute to the scientific significance and clinical relevance of the study's findings.

Results: In my opinion, the authors have thoughtfully presented an extensive table encompassing demographic and clinical data. However, I recommend further elucidating the interpretation of these findings within the text. For instance, why did certain variables not show significant differences between the two groups, and what implications does this have for the study? additionally, I would suggest to clearly present the correlation coefficients and their significance levels when discussing associations between variables to provide readers with a sense of the strength and direction of these relationships.

Discussion: When discussing the study's findings, authors should consider providing a more in-depth comparison with relevant previous studies. How do these results align or differ from previous research? Also, please discuss the clinical implications of these findings. How could the identification of reduced hemoglobin levels and their association with CAD impact patient care and management, particularly in individuals with T2DM?

In according to the previous comment, I would ask the authors to include a proper and defined ‘Limitations and future directions’ section before the end of the manuscript, in which authors can describe in detail and report all the technical issues brought to the surface,

References: Authors should consider revising the bibliography, as there are several incorrect citations. Indeed, according to the Journal’s guidelines, they should provide the abbreviated journal name in italics, the year of publication in bold, the volume number in italics for all the references.

I hope that, after these careful revisions, the manuscript can meet the Journal’s high standards for publication. I am available for a new round of revision of this article. 

Best regards,

Reviewer

References: 

1. https://plos.org/resource/how-to-write-a-great-title/

2. https://www.nature.com/nature-index/news-blog/how-to-write-a-good-research-science-academic-paper-title

3. https://www.indeed.com/career-advice/career-development/catchy-title

4. https://www.mdpi.com/journal/ijms/instructions

5. https://dept.writing.wisc.edu/wac/writing-an-introduction-for-a-scientific-paper/

6. DOI: 10.17219/acem/165944 

7. https://doi.org/10.3390/ijms24065926

8. DOI: 10.3390/biomedicines11030945

9. https://doi.org/10.3389/fnmol.2023.1217090

10. https://doi.org/10.3390/biomedicines11051248

Minor editing of English language required.

Author Response

Dear Reviewer 2.

Thank you for your thorough review and suggestions and for your kind words.

  • I recommend revising the title. The title's structure is quite complex, which might make it harder for readers to quickly grasp the main message, therefore simplifying the structure could enhance readability. Also, in my opinion, in the present form it is quite long, which might discourage some readers from engaging with the article. A potential revised title could be: "Associations Between Inflammation, Hemoglobin Levels, and Coronary Artery Disease in Non-Albuminuric Individuals With and Without Type 2 Diabetes." [1-3].

Thank you for your suggestion and for the references that we will take into account in our future work. We have "adopted" the proposed title because it simply and accurately reflects the main objective of the manuscript. Thanks again.

  • A graphical abstract that will visually summarize the main findings of the manuscript is highly recommended.

We have included a final figure (Figure 2) that summarizes the main findings.

  • Abstract: According to the Journal’s guidelines, this section should be presented as a short summary of about 200 words maximum that objectively represents the article. It should let readers get the gist or essence of the manuscript quickly, prepare the readers to follow the detailed information, analyses, and arguments in the full paper and, most of all, it should help readers remember key points from your paper. Please, consider rewrite this paragraph following these instructions [4].

Thanks for your observation. We have reduced the abstract to 184 words. The new version is:

"In this cross-sectional study we evaluated the associations of inflammation and hemoglobin with coronary artery disease (CAD) in subjects with type 2 diabetes mellitus (T2DM) and preserved kidney function. We recruited 638 participants -254 with T2DM- subjected to coronary angiography with no known cardiovascular disease, normal glomerular filtration rates, and without albuminuria. Hemoglobin and serum levels of inflammatory markers, including high-sensitivity C-reactive protein (hs-CRP), were measured. Multivariable analyses showed that inflammatory markers were not related to the severity of the stenosis in the group of subjects with diabetes. Conversely, inflammatory cytokines and albuminuria were directly related to the percentage of stenosis in subjects without T2DM (R2 = 0.038, P<0.001). Patients with diabetes presented lower hemoglobin levels, particularly in those who also had significant CAD (14.4 [13.6-15.1] vs. 13.6 [12.2-14.8] g/dL, P = 0.03). Similarly, hemoglobin levels and albuminuria were inversely related with the severity of stenosis exclusively in subjects with diabetes, even after adjusting for multiple confounding factors (R2 = 0.081, P<0.001). We conclude that reductions in hemoglobin levels in subjects with T2DM and normoalbuminuria may constitute a more relevant risk factor for CAD than inflammation"

  • Keywords: Please list ten keywords chosen from Medical Subject Headings (MeSH) and use as many as possible in the title and in the first two sentences of the abstract. I would suggest adding “Cardiovascular Risk” as keyword.

We have included "Cardiovascular risk" as a new keyword.

  • Introduction: The authors need to reorganize this section with several paragraphs made up of about 1000 words, introducing information on the main constructs of this study, which should be understood by a reader in any discipline, and making it persuasive enough to put forward the main purpose of the current research the author has conducted and the specific purpose the author has intended by this protocol. I would like to encourage the authors to present the introduction starting with the general background, proceeding to the specific background on the prevalence of T2DM, its association with cardiovascular disease, the complexity of risk stratification, the role of biomarkers, the connection between inflammation and atherosclerosis, the potential role of anemia in cardiovascular risk among T2DM patients. Those main structures should be organized in a logical and cohesive manner [5]. 

We appreciate the reviewer’s suggestions. We have now reorganized the Introduction section according to her/his indications.

  • In this regard, I believe that the Introduction section would benefit from additional information to enhance its clarity and contextualization. To strengthen this section, I suggest incorporating a brief discussion about the potential neural substrates that connect diabetes, inflammation, and cardiovascular risk. Recent research has increasingly recognized the role of neural substrates in mediating the intricate relationship between diabetes, inflammation, and cardiovascular health. Neural pathways, such as the autonomic nervous system and the hypothalamic-pituitary-adrenal axis, play pivotal roles in regulating systemic inflammation, glucose metabolism, and vascular function. The dysregulation of these pathways in diabetes could contribute to chronic inflammation and endothelial dysfunction, key drivers of cardiovascular complications. Exploring the neural underpinnings of the observed associations between reduced hemoglobin levels, inflammatory markers, and CAD in individuals with T2DM could provide valuable insights into the complex interplay of physiological processes involved [6-7]. Addressing this aspect could potentially open avenues for more targeted interventions that focus on modulating neural substrates alongside conventional risk factors. By integrating this discussion of neural substrates, the manuscript would not only underscore the multidimensional nature of the investigated associations but also foster a deeper understanding of the mechanisms at play [8-10]. This addition would further contribute to the scientific significance and clinical relevance of the study's findings.

Following the recommendations of the reviewer, we have included a brief discussion about the potential role of neural substrates in the relationship among diabetes, inflammation and cardiovascular disease in the Introduction section. This states the following:

On the other hand, many different substrates have been proposed as mediators of the intricate relationship between diabetes, inflammation, and cardiovascular risk. In recent years, the role of neural substrates is increasingly recognized and different studies evi-dence pivotal roles of neural pathways, such as the autonomic nervous system and the hypothalamic-pituitary-adrenal axis, in regulating systemic inflammation, glucose me-tabolism, and vascular function [28]. The dysregulation of these pathways in diabetes could contribute to chronic inflammation and endothelial dysfunction, key drivers of car-diovascular complications.

  • Results: In my opinion, the authors have thoughtfully presented an extensive table encompassing demographic and clinical data. However, I recommend further elucidating the interpretation of these findings within the text. For instance, why did certain variables not show significant differences between the two groups, and what implications does this have for the study? additionally, I would suggest to clearly present the correlation coefficients and their significance levels when discussing associations between variables to provide readers with a sense of the strength and direction of these relationships.

We have further interpreted the results obtained with new sentences in the results section. Similarly, we included statistical data results in the discussion of the results. The new text is:

  • “Serum total cholesterol, HDL- and LDL-cholesterol levels were lower in the subjects with T2DM, probably due to the higher percentage of patients treated with lipid lowering medications in this group (23.7% vs. 45.7%, P = 0.03)”.
  • “Fasting glucose and Hb1Ac were higher in the group with T2DM (98 [89-112.8] vs. 148 [133-183.3] mg/dL, P < 0.001; 5.6 ± 0.97 vs. 7.8 ± 0.89 %, P < 0.001, respectively). Similarly, the TyG index also presented significantly higher values in this group of patients (4.77 [4.63-4.93] vs. 4.93 [4.76-5.17], P < 0.001)”
  • “We performed bivariate correlation analyses in each group for determining the associations of hemoglobin levels with different parameters (Table 2). Hemoglobin concentration was inversely related with ACR both in non-T2DM (r = -0.103, P = 0.043) and T2DM patients (r = -0.224, P = 0.005). Hemoglobin was also significantly and inversely associated with fasting glucose (r = -0.199, P = 0.013) and hs-CRP values (r = -0.167, P = 0.038) only in the group of subjects with T2DM. Interestingly, and again exclusively in T2DM patients, hemoglobin was inversely related with the values of the SSI (r = -0.19, P = 0.018), particularly with the percentage of stenosis observed in two major epicardial arteries: RCA (r = -0.167, P = 0.039) and LAD (r = -0.182, P = 0.036) (Table 2)”.
  • “The sub-analysis of the group of subjects with T2DM attending to the occurrence of sig-nificant CAD revealed a higher incidence of anemia in patients with significant CAD (13.1 % vs. 34.2%, P = 0.03) (Table 3). Congruently, the subjects with CAD also presented reduced levels of hemoglobin (14.4 [13.6-15.1] vs. 13.6 [12.2-14.8] g/dL, P = 0.03), together with increased levels of ACR (4.73 [3.25-12.6] vs. 9 [4.4-18.8] mg/g, P = 0.04) and hs-CRP (3.14 [1.3-6.1] vs. 4.6 [2.1-6.8] mg/L, P = 0.02), and higher SBP (P = 0.04). No differences were observed in TyG index values (5 [4.74-5.34] vs. 4.93 [4.77-5.14], P = 0.36), nor in inflammatory cytokines TNFα and IL6 (2 [1.59-2.78] vs. 2.22 [1.59-3.1] pg/mL, P = 0.19; 5.43 [3-8.89] vs. 7.47 (3.51-14) pg/mL, P = 0.21, respectively), nor in NLR values (2.51 [1.75-3.68] vs. 2.19 [1.74-3.38], P = 0.46).
  • Discussion: When discussing the study's findings, authors should consider providing a more in-depth comparison with relevant previous studies. How do these results align or differ from previous research? Also, please discuss the clinical implications of these findings. How could the identification of reduced hemoglobin levels and their association with CAD impact patient care and management, particularly in individuals with T2DM?

The clinical implications and the potential impact of our results are now depicted in the new section “Conclusions and future directions” (see below).

  • In according to the previous comment, I would ask the authors to include a proper and defined ‘Limitations and future directions’ section before the end of the manuscript, in which authors can describe in detail and report all the technical issues brought to the surface.

We have depicted the limitations of the study at the end of the discussion section. Following the suggestions of the reviewer, we included a new section before the end of the manuscript: “Conclusions and future directions”. In this section, we discuss the clinical implications of our findings.

Conclusions and future directions: A clear association between anemia and CAD is present in patients with T2DM and normoalbuminuria. While microalbuminuria in patients with T2DM has increasingly being identified and treated, the screening for anemia should perhaps be extended to this high-risk population independently of ACR values. Our results suggest that early detection and correction of anemia in routine blood tests in this population with early signs of nephropathy can potentially improve the progression of this complication”.

  • References: Authors should consider revising the bibliography, as there are several incorrect citations. Indeed, according to the Journal’s guidelines, they should provide the abbreviated journal name in italics, the year of publication in bold, the volume number in italics for all the references.

We have revised and adjusted the bibliography according to the journal’s guidelines.

Round 2

Reviewer 2 Report

Dear Authors,

I am pleased to acknowledge that you have indeed addressed all of my concerns and queries in a clear and precise manner. Your responses have provided valuable insights into the modifications made to the manuscript in light of my comments. It is evident that you have taken great care to ensure that the revised manuscript aligns more closely with the scientific rigor expected for publication in IJMS.

Upon reviewing the updated version, I find that the inclusion of the additional studies has indeed enriched the understanding of neural substrates in mediating the intricate relationship between diabetes, inflammation, and cardiovascular health. The provided studies contribute significantly to the comprehensiveness of the section. However, in order to provide a more holistic view of the neural structures, I believe there's still an opportunity to expand upon certain factors. Specifically, the discussion of the the neural underpinnings of the observed associations between reduced hemoglobin levels, inflammatory markers, and CAD in individuals with T2DM could offer a deeper insight into the mechanisms at play (https://doi.org/10.3390/biomedicines11030945; https://doi.org/10.1016/j.neubiorev.2023.105163; https://doi.org/10.3390/biomedicines10122999; https://doi.org/10.3389/fpsyt.2023.1225755). This would provide readers with a clearer understanding of the association of T2DM with cardiovascular disease, the complexity of risk stratification, the role of biomarkers.

I want to reiterate my appreciation for your responsiveness and willingness to consider these suggestions. I believe that this minor revision will significantly enhance the quality and impact of the Introduction section. 

Thank you once again for your dedication to improving the manuscript. I look forward to seeing the continued progress.

Best regards,

Reviewer

Author Response

Again, we thank the reviewer for his invaluable feedback. We agree that the potential involvement of neural substrates in mediating the complex relationships between diabetes, inflammation and cardiovascular health, which is the main focus of our work, may be of potential interest to readers for further investigation. Following the reviewer's suggestion, we have expanded in the introduction the discussion of the relationships that could have neural underpinnings in these interactions according to the suggested literature.